# Impact of the Use of a Larger Forearm Artery on the Efficacy and Safety of Transradial and Transulnar Access: A Randomized Trial with Preprocedural Ultrasonography

**DOI:** 10.3390/jcm9113607

**Published:** 2020-11-09

**Authors:** Pawel Lewandowski, Anna Zuk, Tomasz Slomski, Pawel Maciejewski, Bogumil Ramotowski, Andrzej Budaj

**Affiliations:** Cardiology Department, Centre of Postgraduate Medical Education, Grochowski Hospital, 04-073 Warsaw, Poland; azuk@cmkp.edu.pl (A.Z.); tomek.slomski@gmail.com (T.S.); pmaciejewski@cmkp.edu.pl (P.M.); bramotowski@cmkp.edu.pl (B.R.); abudaj@cmkp.edu.pl (A.B.)

**Keywords:** transradial access, transulnar access, complications, ultrasound examination

## Abstract

(1) Background: We aimed to assess the impact of the selection of a larger radial or ulnar artery on the efficacy of access and vascular complications, based on preprocedural ultrasonographic examination. (2) Methods: This prospective, randomized trial included patients undergoing coronary angiography (CAG) or percutaneous coronary intervention (PCI). Patients were randomized into either a larger ulnar artery (UA) or radial artery (RA) group or smaller UA/RA group. The primary endpoint was successful CAG/PCI without crossover to another artery. The secondary endpoints were incidences of radial or ulnar artery occlusion (RAO/UAO) at the 24 h and 30 day follow-up. (3) Results: Between 2017 and 2018, 200 patients (107 men, mean age 68 ± 8 years) were enrolled. The success of CAG/PCI via the access site was 98% and 83% (*p* < 0.001) in the larger UA/RA group and smaller UA/RA group, respectively. The independent factor for CAG/PCI success was the larger artery (OR 9.8, 95%CI 2.11–45.5; *p* < 0.005). The larger UA/RA was superior, with RAO/UAO at 24 h: OR 0.07, 95%CI 0.09–0.61; *p* < 0.016; and RAO/UAO at 30 days: OR 0.25, 95%CI 0.05–0.12; *p* < 0.001. (4) Conclusions: Larger artery access was shown to be more efficient and safer than recessive forearm artery access.

## 1. Introduction

In recent years, the radial artery has been the preferred vascular access point in coronary interventions (transradial access (TRA)) [1,2,3]. The ulnar artery (UA) and radial artery (RA) are the terminal branches of the brachial artery. Transulnar access (TUA) is as effective and safe as TRA, especially in the hands of experienced operators [4,5,6,7]. However, UA is located deeper and is often more difficult to locate and cannulate than the RA [5]. There is considerable variability in the forearm arterial diam. While there are no studies on large groups to assess RA and UA sizes, data from randomized trials have revealed that the smaller diam. of the radial artery or larger sheath size are significant factors of radial artery occlusion (RAO) [8,9]. It seems that in order to limit vascular complications and reduce the incidence of cramps and pain, larger RA or UA should be used. In 2019, the Society of Cardiovascular Angiography and Interventions (SCAI) recommended using preprocedural and real-time ultrasonography to facilitate forearm vascular access [10].

No randomized trials are currently evaluating the UA or RA diam. or the effect that this variable may have on the incidence of complications, quality, and efficacy of vascular access.

Therefore, in the present study, we compared the impact of the use of a larger forearm artery identified with preprocedural ultrasonographic examination (pre-US) on the efficacy of vascular access and the frequency of complications in patients scheduled for elective coronary angiography (CAG) and percutaneous coronary intervention (PCI).

## 2. Materials and Methods

### 2.1. Study Design

This was a prospective, single-center, randomized study, conducted between 2017 and 2018. Patients referred for elective CAG/PCI were included. Regardless of sex, patients aged ≥ 18 years and hospitalized for elective CAG were eligible for this trial. Those who had undergone previous vascular interventions via RA/UA were also included. Exclusion criteria were as follows: under 18 years of age, hemodialysis therapy, nonpalpable pulse over RA/UA, Non-ST-elevation myocardial infarction (NSTEMI), STEMI, coronary artery bypass graft surgery (CABG) with a bilateral left internal mammary artery or right internal mammary artery or forearm arteries, radial or ulnar artery occlusion confirmed by ultrasonographic examination, similar dimension of RA and UA. The clinical trial registration number was DRKS00012923.

Pre-US examinations of the right and left forearm arteries were performed by two experienced sonographers prior to CAG, using the ultrasound scanner EUB 5000 (Hitachi, Ltd., Tokyo, Japan) with a 5–10 MHz linear probe in B-mode.

The average of two measurements of crossectional diam. (perpendicular to each other) was calculated. Measurement of the internal artery diam. was performed at the wrist level (1st diam.), at half of the length of the forearm (2nd diam.), and below the elbow (3rd diam.). The index of artery domination (IxD) was determined by the average of the levels of forearm measurements (Figure 1).

The pre-US examination was applied to identify the larger artery. For inclusion in the study, the difference between patients’ arteries had to be greater than 0.1 mm of IxD. Based on IxD, patients were assigned to UA larger group (group A) or RA larger group (group B). After the pre-US examination, layered randomization was performed using Research Randomizer Software (Version 4.0) (Urbaniak, G.C., & Plous, S. (2013), Lancaster, Pennsylvania PA, USA). In group A, patients were randomly assigned to two subgroups wherein the invasive procedure was performed through the larger UA or the smaller RA. In group B, randomized treatment assignment was for the larger RA or the smaller UA. For statistical analysis, two groups were created: larger artery group (larger UA/RA) and smaller artery (smaller UA/RA). The allocation of patients after randomization is presented in Figure 2 and in Appendix A.

Pre-US examination results were blinded for operators. They were asked to use the radial artery or ulnar artery for vascular access, according to the randomization. Compared to IxD, the sensitivity and specificity of 1st, 2nd and 3rd diam. in the detection of larger artery were evaluated. The right limb was preferred for CAG/PCI, in accordance with current clinical practice. Experienced certified interventional cardiologists, experts in both TRA and TUA procedures, performed the CAG or PCI. Standard 6-Fr radial introducers (Radifocus Introducer II, Terumo Europe, Leuven, Belgium), 6- or 5-Fr diagnostic catheters (Angiodyn, B Braun, Melsungen, Germany), and 5- or 6-Fr guiding catheters (Luncher, Medtronic, Danvers, MA, USA) were used for CAG/PCI. If the primary access failed, the next site of vascular access was the ipsilateral RA or UA. The permitted effective duration of cannulation (time to insert the vascular sheaths and achieve blood outflow) was <15 min for scheduled procedures. After vascular access was established, a bolus of unfractionated heparin (5000 IU) was administered, and angiography of the upper limb circulation was recorded. To prevent vascular spasm, 0.2 mg intra-coronary nitroglycerin bolus was administered. In the case of PCI, the total heparin dose was 70–100 IU/kg.

There are limited data from randomized controlled trials (RCTs) concerning postprocedural artery perforation; for this reason, before the removal of the vascular sheath, upper limb angiography was performed to evaluate this complication. According to standard protocols, CAG was performed along with PCI ad hoc if necessary. Hemostasis was achieved with short manual compression at the beginning and continuous pressure with standard sterile artery gauze compression for at least 2 h afterward.

#### 2.1.1. Efficacy of Vascular Access

The primary endpoint was efficacy. It was defined as a successful CAG/PCI without the crossover to another vascular access during CAG/PCI.

#### 2.1.2. Safety of Vascular Access

The secondary endpoints were the incidence of RAO/UAO at the 24 h and 30 day follow-ups after CAG or PCI.

Large hematoma of the forearm (grade 4 on the EASY scale) [11], ischemic stroke/TIA, major bleeding (type 3 and 5 as per the Bleeding Academic Research Consortium) [12], iatrogenic pseudoaneurysm (IPA), arteriovenous fistula (a-v fistula), and over 50% stenosis events in the used arteries were also recorded at the 24 h and 30 day follow-ups. To detect RAO/UAO and other local complications, an US examination was performed at 24 h and after 30 days of observation by a highly-skilled ultrasonographer. Events of artery perforation visualized in arteriography were recorded additionally.

The study was approved by the Centre of Postgraduate Medical Education Bioethical Committee and performed in accordance with the Declaration of Helsinki. All patients provided informed consent.

### 2.2. Statistical Methods

Patient characteristics are expressed as means and standard deviations (SD) for continuous data and frequency tables for categorical data. The analysis of efficacy was examined on an intention-to-treat basis in all patients who underwent randomization and were assigned to larger or smaller artery groups. The safety endpoint outcomes were assessed using on-treatment analysis and on an intention-to-treat analysis presented in Appendix A. Multiple logistic regression models were fitted to identify independent predictors for binary endpoints at a statistical significance of 5%. Odds ratios (ORs) with 95% confidence intervals (CIs) were calculated for significant predictors. The frequency of complications was compared between groups using the chi^2^ or Fisher exact test. The interobserver variability was assessed using Bland-Altman plots and Lin’s concordance correlation coefficients of absolute agreement. We calculated the mean difference and 95% limits of agreement, defined as the average difference (assumed to be 0 in cases of no consistent bias) ± 1.96 SDs. Values of ρc strength of agreement (Lin’s concordance correlation) <0.90 were considered “poor”, 0.90–0.95 as “moderate”, 0.95–0.99 as “substantial”, and >0.99 as “almost perfect.” Taking into account previous research, it was assumed that the complex safety endpoints would occur in 12.1% of the population (on average 6%) [5]. In order to achieve a statistical significance of 90%, assuming that both trial groups were equal, a sample minimum of *n* = 100 per group was required. Calculations were performed using STATA v.14 software (StataCorp LLC, College Station, TX, USA).

## 3. Results

Two hundred patients (107 men and 93 women, mean age 68 ± 8 years) were randomized to four groups: group A with larger UA (*n* = 50), group A with smaller RA (*n* = 50), group B with larger RA (*n* = 50), and group B with smaller UA (*n* = 50) (Figure 2). Due to crossover between the TRA and TUA, the larger UA/RA and smaller UA/RA groups finally included 115 (58%) patients and 85 (42%) patients, respectively. The baseline characteristics of the patients in the larger and smaller groups are summarized in Table 1. There were no significant differences between the groups.

Right limb arterial access was applied in 101 (88%) of patients in the larger group and in 70 (80%) of patients in the smaller group (*p* = 0.2). Left limb arterial access was chosen in 29 (14.5%) of patients in both groups (larger group *n* = 14 (12%); smaller group *n* = 14 (17%)) (Table 2). Left forearm arterial access was chosen because of the necessity of evaluating the left internal mammary artery (larger group *n* = 4, smaller group *n* = 3), patient preference (larger group *n* = 8, smaller group *n* = 7), and right arm RAO due to prior invasive procedures (larger group *n* = 3, smaller group *n* = 4). Elective PCI was performed in 11 (10%) patients of the larger group and in 3 (4%) patients of the smaller group. In the remaining cases, CAG or CAG with PCI ad hoc was performed (Table 2).

To assess interobserver variability, 10 patients were examined in the same environmental conditions, with the same US scanner, by both observers. Measurements of diam. (two at each level) at three levels of the forearm were taken. In interobserver variability analysis, the value of ρc strength of agreement (Lin’s concordance correlation) between observers was 0.96 for RA and 0.92 for UA.

In group A, the UA was larger than RA at all examined levels, and in group B, the RA was larger than UA (Table 3). The half forearm mean diam. (2nd diam.) with the specificity of 90%, sensitivity of 96% and correct classification of 93% had the highest sensitivity and specificity of three levels mean diam. in the prediction of artery domination.

### 3.1. Efficacy: Intention to Treat Analysis

There were incidents in which it was necessary to change the initial allocation route. The CAG/PCI success of vascular access was observed in 98 (98%) and 83 (83%), (*p* < 0.001) of those in the larger UA/RA and smaller UA/RA groups, respectively, (Table 4).

In the multivariate regression analysis, the independent factors for CAG/PCI success using the larger or smaller artery were as follows: (1) use of larger artery: OR, 9.8; 95% CI, 2.11–45.5; *p* < 0.005) body mass index (BMI) over median: OR, 3.5; 95% CI, 1.09–11.28; *p* < 0.0035; and (2) anatomical abnormalities of UA: OR, 0.09; 95% CI, 0.17–0.48; *p* < 0.005. In all cases of crossover, the second vascular access was successful. In the larger UA/RA group, there was one case of crossover from UA to smaller RA, and one case of crossover from RA to smaller UA. In the smaller UA/RA group, there were eight crossovers from RA to larger UA and nine from UA to RA. The main reasons of crossover are presented in Table 5.

### 3.2. Safety

The incidences of secondary endpoints are presented in Table 6. Only one (0.9%) (RA, *n* = 1; UA, *n* = 0) patient in the larger group experienced RAO/UAO vs. nine (11%) patients (RA, *n* = 9; UA, *n* = 0) in the smaller group 24 h after CAG/PCI (*p* < 0.002). There was a statistically significant difference in the incidence of RAO/UAO between groups at the 30-day follow-up (*p* < 0.001). In the larger UA/RA, there were two (1.7%) RAO/UAO (RA, *n* = 1; UA, *n* = 1), and in the smaller group, there were 13 (15%) cases of occlusion, all in the RA (Table 6). In intention-to-treat analysis, in the larger artery group (*n* = 100), there was a significantly lower rate of RAO/UAO at 24 h and 30-day follow-up. It is presented in the Appendix A.

There were no significant differences between groups in terms of frequency of local complications, except RAO/UAO at the 24 h and 30 day follow-up. Significant stenosis of the artery (over 50% of diam.) was diagnosed in seven (6%) patients of the larger group and 12 (14%) of the smaller group; however, the difference was of borderline statistical significance (*p* < 0.056) (Table 6).

Arterial angiography was performed to observe intra-arterial complications. Perforation of the artery wall occurred in ten (9%) and ten (12%) patients in the larger and smaller groups, respectively. To prevent the growth of hematoma, balloon angioplasty in place of perforation was performed (Figure 3).

The following independent factors for RAO/UAO at the 24 h and 30 day follow-up were identified: access by larger versus smaller artery for intervention, perforation of the artery diagnosed in angiography, and use of TRA versus TUA for intervention, and (Table 7).

## 4. Discussion

This single-center, randomized study showed that the use of the larger forearm artery for CAG/PCI had an impact on the reduction of crossover rates in the larger artery group vs. the smaller artery group, and significantly reduced the frequency of RAO/UAO.

### 4.1. Comparison with Previous Studies

#### 4.1.1. TRA vs. TUA for CAG/PCI

TRA was not superior to TUA when the CAG or PCI was performed by an experienced operator [6]. A meta-analysis by Dahal et al. compared the success rates, efficacy, and safety of TRA and TUA [13]. The data from five randomized clinical trials of 2744 patients were included in this analysis, with nearly an equal number of patients undergoing TRA and TUA catheterizations (1360 and 1384, respectively). The primary outcomes were major adverse cardiac events, and the secondary outcome was the composite endpoint of access-related complications. The authors concluded that TUA resulted in higher rates of site access failure and crossover rate. However, it had similar efficacy, safety, and procedural times to TRA. Fernandez et al., who conducted a meta-analysis of six randomized controlled trials, also supported the use of UA as an alternative to RA for cardiac catheterization [14].

#### 4.1.2. Preprocedural Ultrasonographic Examination

Pre-US assessment can assist in the selection of the larger artery when the use of larger guiding catheters is necessary (i.e., rotablation procedures, endomyocardial biopsy, and double-stent technique). Chugh et al. assessed the feasibility and utility of preprocedural imaging of forearm arteries and found that outcomes (procedure success, reduction of patient discomfort, arterial spasm, and fluoroscopy time) were better in the group with pre-US (*n* = 2244) than in the group without it (*n* = 37,810) [15]. In a recent systematic review and meta-analysis of 12 trials of ultrasound-guided radial access, Moussa Pacha et al. showed that this technique improved the first-attempt success rate (risk ratio [RR] 1.35, 95%CI 1.16–1.57) and decreased the necessity of crossover (RR 0.52, 95%CI 0.32–0.87) [16]. In a recent study evaluating the anatomy and dimensions of forearm arteries in a group of 125 patients, the mean UA and RA diam. were 2.07 ± 0.27 mm and 2.03 ± 0.28 mm, respectively [17]. As early as in 1999, Saito et al. observed an unfavorable relationship in which the diam. of the guiding catheters were too large with respect to the diam. of the RA [18]. To limit vascular complications, primarily arterial occlusion and large hematomas, an artery of a larger diam. should be used. To evaluate the larger forearm artery, we propose to use pre-US examination and IxD, which was based on six measurements of diam. at three different levels of the upper limb. In our analysis, the half-forearm mean diam. had the highest sensitivity and specificity to predict the domination of the forearm artery and could be easier to use in practice.

#### 4.1.3. Efficacy and Safety of TRA and TUA

In our trial, the RAO/UAO rate was 0.9% and 1.7% (*p* < 0.002) in the larger artery group and 11% and 15% (*p* < 0.001) in the smaller group at 24 h and 30 days, respectively. The frequency of the other local complications was similar in both groups (Table 6). In our opinion, the most crucial benefit of the use of the larger artery was the reduction of the frequency of RAO/UAO. The frequency of the other local complications was similar in both groups (Table 6). In our opinion, the most crucial benefit of the use of the larger artery was the reduction of the frequency of RAO/UAO. In earlier studies, the range rate of RAO was from 3.9% to almost 33% [5,8,19,20]. A review and meta-analysis of 66 trials (*n* = 31,345) showed that RAO ranged from less than 1 to 33%, and higher heparin doses (5000 IU) and shorter compression times were effective in reducing the risk of RAO events [21].

RAO/UAO could occur in the forearm at different levels, and not just at the site of cannulation. Our findings suggest that the primary independent factors impacting the frequency of complications are the diam. of the forearm arteries, artery perforation, and the use of TRA (Table 5).

#### 4.1.4. Forearm Artery Perforation after CAG/PCI

Artery perforation is a vascular complication with unknown incidence. Knowledge of this complication is based mostly on case reports and very limited data from TRA/TUA efficacy and safety trials [6,20,22]. Our data suggest that this complication could be diagnosed more often if final angiograms of RA or UA after CAG/PCI were performed. Artery perforation can be the basis of other complications radial artery occlusion, small and large hematoma, bleeding, pseudoaneurysms or a-v fistula. If this complication occurs, it could be managed conservatively with prolonged external compression, balloon angioplasty or the implantation of a stentgraft. Early diagnosis and proper treatment of artery perforation could lead to a reduction of the rate of local complications (Figure 3).

#### 4.1.5. The Use of UA as Vascular Access

Popularizing TUA may have an impact on the reduction of crossovers to TFA when TRA fails or is unavailable. Compared to our findings, previous randomized trials evaluating TRA or TUA for CAG/PCI reported higher crossover rates. One of the largest trials (AURA of ARTEMIS) that tested the noninferiority of TUA to TRA was prematurely terminated because of the high crossover rate in TUA compared to TRA [5]. Bauman et al., in a retrospective analysis, evaluated US-guided cannulation access to the RA or UA, the crossover rate among 1000 consecutive patients (977 TRA; 23 TUA) was 0.3% [23]. Geng et al. reported on 535 consecutive patients who were randomly assigned to the TRA or TUA group. Successful puncture of the target artery was achieved in 95.1% and 91.5% of the patients in the TRA and TUA groups, respectively (*p* > 0.05) [7].

### 4.2. Impact on Daily Practice

The UA can be wider than the RA, but it is also more difficult to palpate. Pre-US could identify the optimal forearm artery and cannulation sites. Pre-US or ultrasound-guided cannulation of forearm arteries can be helpful when the pulse is weak for various reasons, including RAO/UAO, spasm of the artery, and vascular disease. Based on multivariate regression analysis, our study showed several practical results. Using larger forearm artery access (RA or UA) is safer and leads to higher success rates of vascular access: “bigger is better.” Perforation is a minor complication; however, it is associated with a higher risk of RAO/UAO. If TUA is bigger than TRA, it could be the first line vascular access site. According to our findings, higher BMI (over the median) and knowledge of arterial anatomy prior to invasive procedures can impact the efficacy of vascular access.

### 4.3. Limitations

There are some limitations of our study that should be acknowledged. First, we included all patients with a history of CAG or PCI. This could have impacted the outcomes of pre-US; nevertheless, these patients comprised a large part of the populations in catheterization labs. Moreover, the higher doses of UHF among patients after PCI may have impacted the RAO/UAO rate in both groups. We did not use nitroglycerin before pre-US to prevent artery spasm during the examination. We collected information on RAO/UAO only in the last 30 days, but it is known that the number of this complication can increase in the 6 months after CAG/PCI. Finally, the number of patients should be larger. In this sense, we are aware that this study may be underpowered regarding the efficacy endpoint.

## 5. Conclusions

The larger artery access was an independent factor of CAG/PCI success. The UA might not only offer alternative forearm access, but also primary vascular access when the larger artery is used. The use of larger RA or UA vs. smaller artery reduced the rate of local complications, including RAO/UAO.

## Figures and Tables

**Figure 1 jcm-09-03607-f001:**
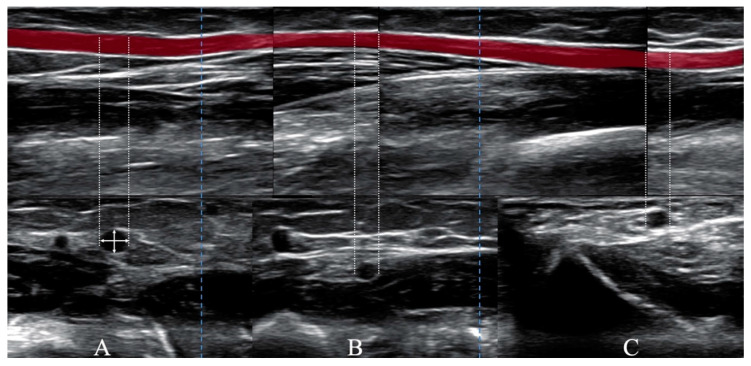
Diagram of radial artery diam. measurements. (**A**) elbow level, (**B**) ½ half forearm level, (**C**) wrist level, crossing arrows in (A)—scheme of measurements of diam. in the transverse projection of the radial artery.

**Figure 2 jcm-09-03607-f002:**
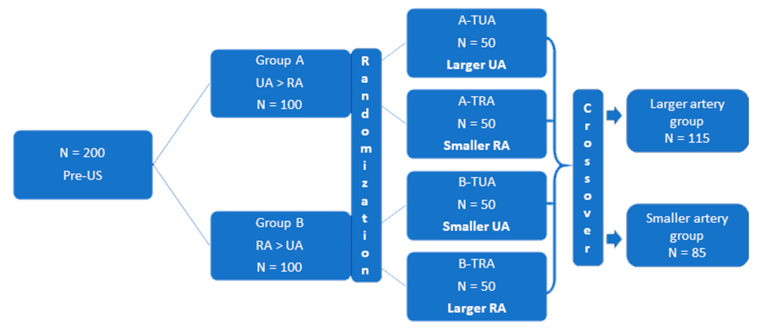
Allocation of patients after layered randomization into larger and smaller artery group. Pre-US—preprocedural ultrasonography, RA—radial artery, UA—ulnar artery, TUA—transulnar access, TRA—transradial access.

**Figure 3 jcm-09-03607-f003:**
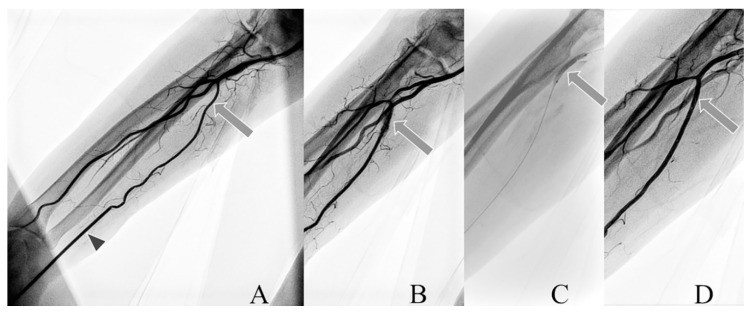
Angiograms of perforated UA. (**A**) vascular sheats (arrowhead), ulnar artery (arrow), (**B**) perforation of the artery (arrow), (**C**) balloon angioplasty (arrow), (**D**) artery without perforation (arrow).

**Table 1 jcm-09-03607-t001:** Demographic and baseline characteristics of the patients.

	Larger UA/RA (*n* = 115)	Smaller UA/RA (*n* = 85)	*p*-Value
Age, years (mean ± SD)	68 ± 8	68.5 ± 7	0.45
Male, *n* (%)	64 (56)	43 (51)	0.5
BMI, kg/m^2^ (mean ± SD)	28.5 ± 4.6	28.5 ± 5.5	0.73
BSA, m^2^ (mean ± SD)	1.95 ± 0.25	1.94 ± 0.2	0.95
**Medical history**			
Smoking, *n* (%)	28 (24)	24 (28)	0.53
Hypertension, *n* (%)	113 (98)	82 (96)	0.65
Hypercholesterolemia, *n* (%)	112 (97)	79 (92)	0.17
Peripheral artery disease, *n* (%)	23 (20)	11 (13)	0.19
Diabetes, *n* (%)	38 (33)	24 (28)	0.46
Stroke, *n* (%)	4 (3)	4 (5)	0.72
Renal insufficiency, *n* (%)	20 (17)	10 (12)	0.27
Myocardial Infarction, *n* (%)	22 (19)	14 (16)	0.62
CABG, *n* (%)	4 (3)	3 (4)	1.0
Prior CAG or PCI, *n* (%)	28 (29)	21 (25)	0.53
**Diagnosis upon admission**			
Suspected CAD, *n* (%)	106 (92)	78 (92)	0.9
CHF, *n* (%)	3 (3)	2 (2)	1.0
Ventricular arrythmia, *n* (%)	6 (5)	5 (6)	0.83
**Preprocedure medications**			
Aspirin, *n* (%)	112 (97)	84 (99)	0.63
Clopidogrel, *n* (%)	110 (96)	82 (96)	1.0
Warfarin, *n* (%)	9 (8)	4 (5)	0.56
NOAC, *n* (%)	6 (5)	6 (7)	0.58
Statin, *n* (%)	107 (93)	80 (94)	0.76
β-Blocker, *n* (%)	107 (93)	77 (91)	0.52

BMI—body mass index, BSA—body surface index, CABG—coronary artery bypass grafting, CAD—coronary artery disease, CAG—coronary angiography, CHF—congestive heart failure, NOAC—Non-Vitamin K antagonist oral anticoagulants PCI—percutaneous coronary intervention, RA—radial artery, UA—ulnar artery.

**Table 2 jcm-09-03607-t002:** Coronary angiography periprocedural characteristics and concomitant medications.

	Larger UA/RA (*n* = 115)	Smaller UA/RA (*n* = 85)	*p*-Value
Angiography alone, *n* (%)	68 (59)	57 (67)	0.25
Angiography and FFR, *n* (%)	5 (4)	1 (1)	0.19
PCI ad hoc, *n* (%)	31 (27)	24 (28)	0.84
Elective PCI, *n* (%)	11 (10)	3 (4)	0.98
TRA, *n* (%)	58 (50.4)	43 (50.6)	0.91
TUA, *n* (%)	57 (49.6)	42 (50.4)	0.98
Right radial or ulnar access, *n* (%)	101 (88)	70 (82)	0.27
Left radial or ulnar access, *n* (%)	14 (12)	15 (18)	0.27
Fluoroscopy time (min) (mean ± SD)	5.4 ± 5.2	4.9 ± 4.2	0.49
Contrast medium (mL) (mean ± SD)	149 ± 94	143 ± 41	0.63
Radiation dose of X-ray (mSv) (mean ± SD)	281 ± 281	246 ± 197	0.64
Time of compression, (min) (mean ± SD)	147 ± 31	149 ± 33	0.49
Nitroglycerin (dose 200 ug) ia, *n* (%)	115 (100)	82 (96)	1.0
Dose of heparin (IU) (mean ± SD)	6008 ± 1600	5900 ± 1544	0.6
**Arterial sheath size**			
6-Fr, *n* (%)	115 (100)	85 (100)	1.0
**Diagnostic catheter size**			
6-Fr, *n* (%) 5-Fr, *n* (%)	105 (91) 2 (1.8)	80 (94) 2 (2)	0.45
**Catheter used for PCI, *n* (%)**	47 (40)	32 (37)	
6-Fr, *n* (%)	47 (100)	32 (100)	1.0

FFR—fractional flow reserved, PCI—percutaneous coronary intervention, RA—radial artery, TRA—transradial access, TUA—transulnar access, UA—ulnar artery.

**Table 3 jcm-09-03607-t003:** Artery diam. and index of domination (IxD) in the pre-US.

	Group A (UA > RA), *n* = 100		Group B (RA > UA), *n* = 100	
	UA	RA	RA	UA
1st diam. (mm) (mean ± SD)	2.3 ± 0.4	1.9 ± 0.3	2.3 ± 0.4	1.8 ± 0.3
2nd diam. (mm) (mean ± SD)	2.5 ± 0.4	2.0 ± 0.4	2.4 ± 0.4	1.9 ± 0.3
3rd diam. (mm) (mean ± SD)	3.0 ± 0.5	2.2 ± 0.4	2.7 ± 0.5	2.4 ± 0.6
Index IxD (mm) (mean ± SD)	2.6 ± 0.4	2.0 ± 0.3	2.5 ± 0.3	2.0 ± 0.3
PSV (cm/s) (mean ± SD)	44 ± 11	40 ± 11	40 ± 12	39 ± 11
EDV (cm/s) (mean ± SD)	6 ± 7	6 ± 8	6 ± 6	5 ± 6
Anatomical abnormalities * *n* (%)	3(3)	8(8)	6(6)	7(7)

EDV—end-diastolic velocity, IxD—index of artery domination, PSV—peak systolic velocity, RA—radial artery, UA—ulnar artery, *—loop or kinking of the artery.

**Table 4 jcm-09-03607-t004:** Primary endpoint—efficacy of vascular access in larger vs. smaller group.

Parameters	Larger UA/RA *n* = 100	Smaller UA/RA *n* = 100	*p*-Value
CAG/PCI success *n* (%)	98 (98)	83 (83)	0.001
Necessity of crossover *n* (%)	2 (2)	17 (17)	0.001
Necessity of crossover *n* (%)	2 (2)	17 (17)	0.001

CAG—coronary angiography, PCI—percutaneous coronary intervention, RA—radial artery, UA—ulnar artery.

**Table 5 jcm-09-03607-t005:** The main reasons of crossover in the larger and smaller group.

Parameters	Larger UA/RA *n* = 100	Smaller UA/RA *n* = 100	*p*-Value
Vessel spasm, *n* (%)	0	5 (5)	0.059
Impalpable pulse, *n* (%)	0	3 (3)	0.246
Prolonged procedure, *n* (%)	2 (2)	4 (4)	0.683
Painful procedure, *n* (%)	0	1 (1)	0.9
No blood outflow after puncture, *n* (%)	0	0	1.0
Impossible artery puncture, *n* (%)	1 (1)	5 (5)	0.2
Impossible wire insertion into artery, *n* (%)	1 (1)	11 (11)	0.005

RA—radial artery, UA—ulnar artery.

**Table 6 jcm-09-03607-t006:** Number (%) of patients with secondary endpoints and complications after CAG/PCI at 24 h and 30 day follow-up.

24 h Follow-Up	Larger UA/RA (*n* = 115)	Smaller UA/RA (*n* = 85)	*p*-Value
RAO/UAO, *n* (%) ^†^	1 (0.9)	9 (11)	0.002 *
Hematoma (grade 4 in EASY scale), *n* (%)	4 (3.5)	3 (3.5)	1.0
Stroke/TIA, *n* (%)	0	1 (1)	0.2 *
Major bleeding, *n* (%)	0	0	
IPA, *n* (%)	1 (1)	3 (4)	0.3 **
a-v fistula, *n* (%)	2 (2)	2 (3)	1.0 **
Significant stenosis of used artery, *n* (%)	5 (4)	8 (9)	0.2 **
**Intra-arterial complications**			
Perforation of artery in angiography, *n* (%)	10 (9)	10 (12)	0.47
**30 day follow-up**			
RAO/UAO, *n* (%) ^†^	2 (1.7)	13 (15)	0.001 **
Hematoma (grade 4 in EASY scale), *n* (%)	3 (2.6)	1 (1.2)	0.5
Stroke/TIA, *n* (%)	0	0	
Major bleeding, *n* (%)	0	0	
IPA, *n* (%)	0	0	
a-v fistula, *n* (%)	0	1	1.0
Significant stenosis of used artery, *n* (%)	7 (6)	12 (14)	0.056 *

a-v fistula—arteriovenous fistula, CAG—coronary angiography, IPA—iatrogenic pseudoaneurysms, PCI—percutaneous coronary intervention, RA—radial artery, RAO—radial artery occlusion, UA—ulnar artery, UAO—ulnar artery occlusion, TIA—transient ischemic attack, ^†^—secondary safety endpoint, *—chi^2^ test, **—Fisher exact test.

**Table 7 jcm-09-03607-t007:** Independent factors for RAO/UAO at 24 h and 30 day follow-ups based on logistic regression calculation.

24 h Follow-Up	OR (95% CI)	*p*-Value
Larger UA/RA	0.07 (0.09–0.61)	0.016
Perforation of the artery	7.24 (1.68–31.05)	0.008
**24 h 30 day follow-up**		
Larger UA/RA	0.025 (0.05–0.12)	0.001
Perforation of the artery	10.38 (2.46–43.68)	0.001
Use of TRA	9.05 (1.75–46.85)	0.009

RA—radial artery, RAO—radial artery occlusion, TRA—transradial artery access, UA—ulnar artery, UAO—ulnar artery occlusion.

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
