# Peer review of "Impact of the Use of a Larger Forearm Artery on the Efficacy and Safety of Transradial and Transulnar Access: A Randomized Trial with Preprocedural Ultrasonography"

_jcm, 2020, doi:10.3390/jcm9113607_

Round 1
Reviewer 1 Report
Highly unique study which intentionally randomized 200 patients to the dominant (larger) or nondominant (smaller) forearm artery. This is impressive and brave given what we know about smaller arteries being more often associated with spasm and radial artery occlusion. The data in this study are consistent with expections and even in a relatively small population show a significant reduction in RAO with using the dominant artery. I found the article's finding highly relevant and would plan on including in presentations on the topic.
Areas where the manuscript could be improved:
1) Please describe in more detail how the operators were blinded to the ultrasound findings
2) Please describe whether ultrasound guidance was used to assist the access
3) Please describe is there was any requirements or instructions for the operators before crossover to the other site (number of attempts, time?)
4) It seems unusual that of the 200 patients exactly half had RA>UA while exactly half had UA>RA in terms of size. (possible given that UA and RA are of similar size, but UA is more often larger than RA). Please confirm that the 200 patients were not selected from some larger population?
5) The analysis of outcomes might benefit from including both an 'as treated' model as the authors present, as well as (perhaps in a supplement) an 'intention to treat' model whereby all patients are analyzed according to their original randomization arm, and crossovers are either excluded or included.
6) The authors should describe how they measured/ascertained RAO/UAO at 24 hrs or 30 days. Did they use ultrasound in all patients?
Reviewer 2 Report
Primary endpoint was success of CAG/PCI as expressed by low crossover rates. The study design is based on the hypothesis (as seen in introduction) that there will be fewer complications, cramps and pain when a larger vessel is used. Although the term dominant artery in the clinical setting of CAG is not popular, it is already known that under an experienced operator UA use can be feasible and safe and that large-diameter RA/UAs are safer and less prone to obstruction.
Regarding secondary endpoints, RAO and UAO seem to be fewer in the dominant arm. Higher UFH doses are already independently associated with lower RAO incidence after CAD without ad-hoc PCI. Thus, the fact that both CAG and PCI procedures were perfomed in this study and that enhanced doses of anticoagulants were used in PCI, the association of larger-diameter vessel and low AO, albeit true, cannot be assessed and established.
Study design lacks efficient power analysis. 200 should be randomized in both arms, however, 115 vs 85 were analyzed. Was that initially considered as an acceptable "drop-out" rate and calculated into the study power? If patients were allocated to the other arm after randomization and arterial access cross-over, this is considered primary endpoint failure. Furthermore, in the primary endpoint analysis 100 patients are analysed in each arm. Which is the case?
Last, regarding complications, with the exception of AO (which is discussed above), they seem to be similar between the two groups. That leans towards non-inferiority.
Round 2
Reviewer 2 Report
Congratulations on your revised manuscript.
I believe you addressed all issues raised in the previous report in a concise and sufficient way.